# Route Optimization for Active Sonar in Underwater Surveillance

**DOI:** 10.3390/s25134139

**Published:** 2025-07-02

**Authors:** Mehmet Gokhan Metin, Mumtaz Karatas, Serol Bulkan

**Affiliations:** 1Department of Industrial Engineering, Marmara University, Istanbul 34854, Türkiye; sbulkan@marmara.edu.tr; 2Biomedical, Industrial and Human Factors Engineering Department, Wright State University, Dayton, OH 45435, USA; mumtaz.karatas@wright.edu

**Keywords:** multistatic sonar, vehicle routing, coverage path planning, hexagonal grid decomposition, ant colony optimization

## Abstract

Multistatic sonar networks (MSNs) have emerged as a powerful approach for enhancing underwater surveillance capabilities. Different from monostatic sonar systems which use collocated sources and receivers, MSNs consist of spatially distributed and independent sources and receivers. In this work, we address the problem of determining the optimal route for a mobile multistatic active sonar source to maximize area coverage, assuming all receiver locations are known in advance. For this purpose, we first develop a Mixed Integer Linear Program (MILP) formulation that determines the route for a single source within a field discretized using a hexagonal grid structure. Next, we propose an Ant Colony Optimization (ACO) heuristic to efficiently solve large problem instances. We perform a series of numerical experiments and compare the performance of the exact MILP solution with that of the proposed ACO heuristic.

## 1. Introduction

Effective deployment of underwater sensors for target detection, localization, and surveillance is the key in underwater surveillance operations. Utilization of passive receivers has been common in the past, though today’s submarine technologies provide more quietness, and modern submarines are nonetheless challenging to detect with passive sensors alone.

Nowadays, multistatic sonar networks (MSNs) have become an inspiring option against completely passive sensor fields and monostatic systems. The concept of multistatic sonar dates back to the 1950s, but it becomes popular during the 1980s. Advances in digital signal processing and underwater communications facilitated the implementation of these networks. Notably, Low-Frequency Active Sonar (LFAS) played a crucial role in achieving extended detection ranges.

A classical monostatic sonar system consists of an active sonar emitter (Transmitter-Tx) and a signal-receiving unit (Receiver-Rx) where the transmitter radiates sound waves that hit an objective and echo back, and the receiver observes the echoed pulses. The distance to the object is identified by the phase discrepancy between sound energy emission and reception. The components of the monostatic sonar system are collocated and may use the same transducer, as shown in Figure 1a.

MSN comprises a network of sonar sensors, both transmitters and receivers, distributed across various platforms. These platforms may include frigates, autonomous underwater vehicles (AUVs), and other appropriate vessels. Since this system utilizes multiple non-collocated transmitters and receivers, and they may vary in number, an emitted sound pulse from any source may be received by any receiver to localize targets. The source may be hull-mounted or towed sonar arrays or a helicopter with a dipped sonar. The receiving unit may be a non-active component of a sonar system placed on a vessel’s hull, part of a static hydrophone structure, or a passive sonobuoy. The basic structures of MSNs are depicted in Figure 1b [1].

The use of MSNs in underwater surveillance is garnering attention as a means to counter underwater enemy targets. Since transmitters and receivers are not collocated, forming a countermeasure operation is complicated for hostile forces. MSNs also present a cost-effective solution for operation planners as an important benefit. Active sources are more expensive than passive receivers due to their greater technical complexity [1,2]. Since MSNs can be formed by utilizing more passive receivers than sources, they can achieve the same sensing capability as a monostatic system at a reduced cost [3]. Also, another advantage of MSNs is to decrease the level of underwater noise contamination by employing a smaller number of sources [4]. Due to these and other factors not detailed here, MSNs are regarded as an effective option for underwater surveillance operations [5].

The coverage of an active sonar source in a bistatic or multistatic configuration is determined by the relative geometry of the transmitter and one or more receivers. It is characterized by a collection of Cassini ovals, each defined by a transmitter–receiver pair. A Cassini oval is the set of points where the product of the distances to two fixed points—the transmitter and receiver—is constant. This geometric shape naturally models the detection zones in bistatic sonar systems, where detection depends on the combined travel paths of the signal from the transmitter to the target and then to the receiver. Unlike the circular coverage zones of monostatic systems, these ovals can be elongated, figure-eight-shaped, or even disconnected, depending on the separation between the source and the receiver. The mathematical complexity of Cassini ovals reflects the challenge in accurately computing sonar coverage in multistatic geometries, but they provide a more realistic representation of how detection probability varies across space.

Due to their unconventional coverage zones, the overall sensing region of an MSN depends on careful placement and use of sensors in an operational area [6]. In this work, our objective is to determine the optimal route for an active mobile platform where several fixed passive receivers are placed and their locations are known beforehand. For this purpose, we first develop a Mixed Integer Linear Program (MILP) formulation which aims to maximize the area coverage in an MSN area search scenario. Next, to solve the problem, we propose an Ant Colony Optimization (ACO)-based heuristic methodology. We perform numerical experiments to compare the performance of the exact and the heuristic approaches on different-size problems.

The remainder of this paper is organized as follows: In Section 2, we review related work on MSN planning and the coverage path planning (CPP) problem. Assumptions and preliminaries for bistatic detection, Cassini ovals and multistatic detection theory are given in Section 3. We describe the mathematical model in Section 4. Section 5 contains the details of our ACO-based heuristic solution. The numerical results and a comparison of our solution with the mathematical model’s performance in terms of solution quality and CPU time are given in Section 6. Finally, in Section 7, we present our conclusions.

## 2. Literature Review

In their work, Washburn and Karatas [7] developed an analytic theory in order to predict the probability of target detection for randomly deployed multistatic fields and further presented pattern optimization, cost efficiency, and reliability-related results for multistatic fields. They also investigated the case of a single source moving along a straight line in a field of receivers and derived a closed-form equation for the sweep width of the MSN. Later, Karatas, Craparo, and Akman [8] studied the utilization of bistatic sonobuoys for the detection of stationary and moving targets. To improve the detection probability of a target, they derived analytic expressions in order to optimize the separation distances between the source and the receiver by using integral geometry and geometric probability concepts. They also validated the accuracy and efficiency of their results by employing Monte Carlo simulations.

Craparo, Fügenschuh, Hof, and Karatas [9] addressed sensor deployment optimization in a multistatic sonar system for tracking a set of targets with predefined locations. They proposed two ILP models for locating sources and receivers in an area that is formed by a discretized set of locations. They also developed a two-phase heuristic procedure to solve the location problem. Craparo and Karatas [3] considered the multistatic source localization problem for an application of position coverage in a multistatic sonar network where targets and receiving units are assumed stationary. Given a predefined sensor range, they developed exact methods and estimation procedures to address this problem and analyze the performance of their solution methods. Their solution procedure starts with identifying a group of possible optimal source locations, followed by a selection phase from these promising candidate locations. They also applied this model to area coverage and barrier coverage applications, which are other types of sensing coverage models by discretizing the operational sector into several targets.

Fügenschuh, Craparo, Karatas, and Buttrey [10] proposed various ILP formulations for the multistatic sensor location problem with coverage maximization and cost minimization objectives.

Considering that the main objective in our problem definition is area coverage maximization, our problem can also be regarded as a CPP problem for an MSN. Galceran and Carreras [11] define the CPP problem as “Finding a path that avoids barriers and covers every point in a volume or region of interest is known as coverage path planning (CPP)”.

Since the CPP problem pertains to the covering salesman problem, which is an altered version of the traveling salesman problem, it is known to be NP-hard; thus, an optimum solution for CPP for large data sets cannot be found in a reasonable time [12]. CPP has been studied by many researchers, and many heuristic- or metaheuristic-based algorithms have been proposed for developing a near-optimum solution where specific types of circumstances and constraints are analyzed such as ACO or genetic algorithms (GAs). It is good practice to discretize the searching area into discrete subareas which do not intersect with each other with the help of a decomposition technique while employing a heuristic- or metaheuristic-based solving algorithm. These discrete sections are known as cells. These cells typically represent just one point along the anticipated path and are proportional to the range of the system that is employed for tracking or covering such as a multistatic transmitter’s range for an underwater coverage mission. The dimensions of these cells depend on the applied technique, and an appropriate approach should be used to ensure total coverage. The discretization approach has a straight performance ramification on solution technique for developing a solution for CPP. The sequence in which those subareas will be covered is decided according to the utilized decomposition process.

Cellular and grid-based discretization approaches are the two techniques for segmenting the area of interest [11,13,14]. The cellular discretization approach segregates the search zone by taking into consideration the features of the area and the platform’s constraints, such as fuel consumption and speed [15,16,17]. Despite the fact that the cellular discretization approach is much plainer, as it takes the features of the search region and the platform in order to identify the sub-vicinities for covering, owing to the numerous constraints, it is challenging to establish the decomposition benchmark. The grid-based discretization approach employs a specified entity to divide the area of interest into manageable portions. The area is decomposed using the grid-based discretization technique, which makes the process simple but adds complexity because it divides the zone without taking the features of the region or the platform’s capabilities into account.

There are studies based on grid-based decomposition applications where the grid cell form is specified as a square. For example, Miao, Lee, and Kang [18] proposed an adaptable CPP approach for large areas that employs a rectangular sub-map decomposition method. They developed this solution for cleaning robots that operate in a large environment where many obstacles exist. They provided a notable reduction in CPP execution time. In another study, Yang, Wang, and Yi [19] considered sensor network optimization problem for improving tracking operations for multistatic radar systems. They established an optimization model for the radar network to enhance area coverage by taking the physical radar features, such as signal transmission range, as constraints. They employed this model with a node clustering network architecture on a discretized surveillance area as a squared grid.

The position and quantity of nearby nodes that the platform can perform maneuvering from in each grid cell differ based on the specified form, which could lead to varied navigational courses. In addition, the grid cell’s side count affects how many directions of movement are possible while determining the search path. Although the problem’s computing complexity expands, this makes it possible to investigate all possible search paths effectively. In this context, the hexagonal-grid-based decomposition approach is promising.

Cho, Park, Park, and Kim [20] proposed a MILP model for minimizing the mission execution time based on a hexagonal-grid-based decomposition method. They studied an algorithm that regards mission execution time as a function of both the size of the reconnaissance zone and several Unmanned Aerial Vehicle (UAV) courses for different reconnaissance zones. Cho, Park, Lee, Shim, and Kim [21] investigated CPP for utilizing UAVs in maritime rescue operations and developed a two-stage solving procedure. They studied a grid-based region discretization to form a graph that consists of nodes and edges and minimized the decomposed region by implementing a rotation of the search area as its dimensions determine execution duration in the first stage. In the second stage, they proposed a MILP model for acquiring an optimal trail which reduces the accomplishment time. They also developed a Randomized Search Heuristic (RSH) procedure to clarify an immense number of instances.

As previously mentioned in Section 1, we consider a multistatic source routing problem in a field of stationary receivers with the objective of maximizing area coverage. This is an attractive ability for naval operations planners. To the best of our knowledge, this is the first study that considers the problem of route optimization for a multistatic source in an underwater surveillance scenario. In order to reach this goal, we propose a mathematical model and ACO-based heuristic solution based on hexagonal-grid-based segmentation. We also present a performance comparison on different problem instances.

## 3. Preliminaries

In this section, we review the preliminaries for bistatic detection, Cassini ovals, and multistatic detection theory, which constitute the backbone of our work.

### 3.1. Bistatic Detection

In a bistatic sonar system, detection happens when the sound waves emitted by the source reflect off a target, creating an echo whose acoustic energy surpasses the receiver’s detection threshold. This threshold, referred to as TH, is affected by the receiver’s sensitivity, environmental conditions, and the configurations for detection and false alarms. As noted in [22], detection in a bistatic system occurs if(1)$$SL-TL_{1}-TL_{2}\geq TH$$

In this framework, SL signifies the source level, whereas $TL_{1}$ and $TL_{2}$ refer to the transmission loss values from the source to the target and from the target to the receiver, respectively, expressed in decibels. Assuming a uniform environment where the signal spreads out in a spherical manner, the transmission loss between two points in the field follows a simple power law. As a result, for some constant m>0, we can rephrase Equation (1) as follows:(2)$$SL-m\log(R_{1})-m\log(R_{2})\geq TH$$
where $R_{1}$ and $R_{2}$ denote the distances from the source to the target and from the target to the receiver, respectively [23]. A common practice is to express the equivalent monostatic detection range as $\rho =\sqrt{R_{1}R_{2}}$, which represents the geometric mean of the two distances. This metric indicates the effectiveness of a bistatic system when both the source and receiver occupy the same location [7,24]. By examining Equation (2), we determine that detection occurs if(3)$$R_{1}R_{2}=\rho^{2}\leq 10^{\frac{1}{m}\left(SL-TH\right)}$$

The inequality given in Equation (3) represents the interior region of a Cassini oval, which serves as the detection zone in a bistatic system [25]. Next, we will outline some fundamental properties of Cassini ovals.

### 3.2. Cassini Ovals

Cassini ovals were introduced by Giovanni Domenico Cassini while developing a systematic explanation for the sun’s trajectory around the world [26]. Many scientific branches have employed Cassini ovals such as acoustics and nuclear physics [27].

A Cassini oval can be defined as the shape formed by the connection of the triangle vertices where the product of the lengths of two adjacent sides and the length of the opposite side is constant [28]. Consider the bistatic triangle in Figure 2.

Let the red vertex denote the top vertex of the triangle; the adjacent sides of the top vertex are symbolized by $R_{1}$ and $R_{2}$, and their product is constant, $\rho^{2}$. The length of the opposite side of the top vertex is also constant, 2a. Therefore, the parameter a can be defined as half of this constant value.

If the system units are located at the points of ±a,0, then the Cassini ovals are determined according to the following formula:(4)x−a2+y2x+a2+y2=ρ4,   a,ρ∈ℝ

The oval’s shape is determined by the ratio of a to ρ and is symmetric corresponding to both the x- and y-axes. The Cassini ovals are quantified according to four conditions:

For the inequality of $a/\rho \leq \sqrt{2}/2$, the oval forms a singular curve similar to an ellipse and crosses the x-axis at $x=\pm \sqrt{a^{2}+\rho^{2}}$.For the interval of $\sqrt{2}/2<a/\rho <1$, the curve has indentation on both the top and bottom sides.For the equality of a/ρ=1, the oval takes the shape of the lemniscate of Bernoulli.And finally, for the inequality of a/ρ>1, the curve is separated into two ovals, and two new x-intercepts take place at $x=\pm \sqrt{a^{2}-\rho^{2}}$.

Cassini ovals are depicted in Figure 3 with different a/ρ values, where ρ is equal to 1.

### 3.3. Multistatic Detection Theory

Since a multistatic sonar system consists of more than one transmitter and receiver, which are separated by some distance, each source and receiver pair forms a bistatic system.

In a monostatic sonar system where the transmitter and receiver are collocated, the $R_{1}$ and $R_{2}$ distances are equal, so the inequality of $R_{1}R_{2}\leq \rho^{2}$ forms a spherical detection zone with a radius of ρ, as shown in Figure 4a. But in a multistatic sonar system which has spatially distributed and independent sources and receivers, the $R_{1}$ and $R_{2}$ have different values, so the inequality of $R_{1}R_{2}\leq \rho^{2}$ forms the interior area of a Cassini oval as explained previously (see Figure 4b).

In this study, we employ the cookie-cutter sensor or definite range model. In this model, if this inequality holds, it is assumed that the target is detected with a probability of 1. Otherwise, it is not detected with probability of 0.

More formally, for the cookie-cutter sensor model, let H be the set of hexagonal nodes in a grid; given a pair of a source which is located at node i∈H and a receiver which is located at node j∈H, a node to be covered k∈H has a binary detection probability $p_{i,j,k}\in \left\{0,1\right\}$. Let $d_{i,j}$ stand for the Euclidean distance between the centers of nodes i and j. It is covered if the node k’s center coordinate is located within the Cassini oval, which is expressed by $d_{i,k}d_{k,j}\leq \rho^{2}$, $p_{i,j,k}=1$; otherwise, no detection occurs, $p_{i,j,k}=0$ [9].

## 4. Problem Definition

### 4.1. Basic Assumptions

Assume that a field of interest is discretized into hexagonal nodes indexed as g and g′ in G. Let r in R represent the set of receivers in this field and $r^{*}$ be the closest receiver to the node g. The distance between g and g′ is denoted as $d_{g,g^{\prime }}$, and the distance between g and $r^{*}$ is represented as $d_{g,r^{*}}$. Also, let ρ represent the range of the day value.

The aim of this work is to determine an optimal route for a mobile multistatic source in order to maximize area coverage within the discrete set of nodes G and the set of receivers R. We list our main assumptions below:

Receivers are stationary, and their locations are known in advance. This situation matches an operational scenario where cheap passive receivers are deployed by a naval aviation platform in the preparation phase of an underwater surveillance operation.The detection range or “range of the day” is known and fixed in all regions of interest. In other words, the environmental conditions are uniform in the region.The search area is discretized as a hexagonal-grid-based region. We further assume that the center coordinates of each hexagonal node in the grid contain a virtual target, and coverage calculations are performed accordingly. If this point lies within the coverage zone of a bistatic pair, then we assume that that particular hexagonal node is fully covered.

Since a hexagonal node can be covered by more than one source–receiver pair, the coverage status of g can be stated as a function of $d_{g,g^{\prime }}$, and $d_{g,r^{*}}$. This function is presented below:(5)$$p_{g}=\left\{\begin{array}{c} 1,\;if\;d_{g,g^{\prime }}d_{g,r^{*}}\leq \rho^{2}\\ 0,\;\mathrm{otherwise}\;\;\;\;\;\; \end{array}\right.$$
where $p_{g}$ is the coverage status of node g. This function states that if node g is within the inner area of the Cassini oval, which is formed by the inequality of $d_{g,g^{\prime }}d_{g,r^{*}}\leq \rho^{2}$, then it is assumed to be covered; otherwise, it is assumed not to be. The coverage statuses of all the nodes in G are combined in a matrix structure, which is represented by $C\left(g,g^{\prime }\right)$, and used in the mathematical model.

### 4.2. Mathematical Model Formulation

Our mathematical model for area coverage maximization for an MSN is based on the MILP model. The sets, indices, parameters, variables, and constraints used in the mathematical model are presented below.


*Sets and Indices*


$g,g^{\prime },g^{\prime \prime }\in G$: Set of nodes.

$N_{g}\subset G$: Subset of adjacent nodes of grid *g*.


*Parameters*


$g^{ss},g^{se}$: Starting and ending nodes of the source.

n: Maximum number of nodes that can be traveled by the source.

$C\left(g,g^{\prime }\right)$: Coverage matrix of hexagonal grid.


*Binary Variables*



$$x_{g,g^{\prime }}=\left\{\begin{array}{c} 1,\;\mathrm{if}\;\mathrm{source}\;\mathrm{travels}\;\mathrm{from}\;\mathrm{node}\;g\;\mathrm{to}\;\mathrm{node}\;g^{\prime }\;\;\;\;\;\;\;\;\;\;\;\;\;\;\;\;\;\;\;\;\;\;\;\;\;\;\;\;\\ 0,\;\mathrm{otherwise}\;\;\;\;\;\;\;\;\;\;\;\;\;\;\;\;\;\;\;\;\;\;\;\;\;\;\;\;\;\;\;\;\;\;\;\;\;\;\;\;\;\;\;\;\;\;\;\;\;\;\;\;\;\;\;\;\; \end{array}\right.$$



$$z_{g}=\left\{\begin{array}{c} 1,\;\mathrm{if}\;\mathrm{source}\;\mathrm{visits}\;\mathrm{node}\;g\;\;\;\;\;\;\;\;\;\;\;\;\;\;\;\;\;\;\;\;\;\;\;\;\\ 0,\;\mathrm{otherwise}\;\;\;\;\;\;\;\;\;\;\;\;\;\;\;\;\;\;\;\;\;\;\;\;\;\;\;\;\;\;\;\;\;\;\;\; \end{array}\right.$$



$$y_{g}=\left\{\begin{array}{c} 1,\;\mathrm{if}\;\mathrm{node}\;g\;\mathrm{is}\;\mathrm{covered}\;\;\;\;\;\;\;\;\;\;\;\;\;\;\;\;\;\;\\ 0,\;\mathrm{otherwise}\;\;\;\;\;\;\;\;\;\;\;\;\;\;\;\;\;\;\;\;\;\;\;\;\;\;\;\; \end{array}\right.$$



$$u_{g}=\mathrm{Technical}\;\mathrm{variable}\;\mathrm{for}\;\mathrm{subtour}\;\mathrm{elimination}$$



*Formulation*



(6)
$$\max\underset{g\in G}{\sum }y_{g}$$



(7)
$$\underset{g^{\prime }\in \left\{G/N_{g}\right\}}{\sum }x_{g,g^{\prime }}=0,\;\;g,g^{\prime }\in G$$



(8)
$$\underset{g\in G}{\sum }x_{g,g^{\prime }}-\underset{g^{\prime \prime }\in G}{\sum }x_{g^{\prime },g^{\prime \prime }}=0,\;\;g^{\prime }\in G,g^{\prime }\neq g^{ss},g^{\prime }\neq g^{se}$$



(9)
$$\underset{g^{\prime }\in G}{\sum }x_{g,g^{\prime }}=1,\;\;g\in G,g=g^{ss}$$



(10)
$$\underset{g\in G}{\sum }x_{g,g^{\prime }}=0,\;\;g^{\prime }\in G,g\prime =g^{ss}$$



(11)
$$\underset{g\in G}{\sum }x_{g,g^{\prime }}=1,\;\;g^{\prime }\in G,g\prime =g^{se}$$



(12)
$$\underset{g^{\prime }\in G}{\sum }x_{g,g^{\prime }}=0,\;\;g\in G,g=g^{se}$$



(13)
$$\underset{g\in G}{\sum }z_{g}\leq n$$



(14)
$$y_{g^{\prime }}\leq \underset{g\in G}{\sum }C\left(g,g^{\prime }\right)*z_{g},\;\;g\in G$$



(15)
$$x_{g,g^{\prime }}\leq z_{g^{\prime }},\;\;g\in G,g^{\prime }\in G$$



(16)
$$\sum z_{g^{\prime }}\leq \underset{g\in G}{\sum }x_{g,g^{\prime }},\;\;g^{\prime }\in G,g^{\prime }\neq g^{ss}$$



(17)
$$z_{g}=1,\;\;g\in G,g=g^{ss}$$



(18)
$$u_{g}-u_{g^{\prime }}+n*x_{g,g^{\prime }}\leq n-1,\;\;g\in G,g^{\prime }\in G,g\neq g^{\prime }$$


The objective Function (6) indicates the overall number of nodes covered. Constraint (7) ensures that there is no transition of source from node g to non-adjacent nodes of g. Constraint (8) ensures that if there is a transition of the source from node g to node $g^{\prime }$, an exit from $g^{\prime }$ has to be made. Constraints (9) and (10) force the source to start from its starting node $g^{ss}$ and ensure that it cannot enter its starting node $g^{ss}$. Constraints (11) and (12) state that the source has to finish at its ending node $g^{se}$ and cannot exit from its ending node $g^{se}$. Constraint (13) is a capacity constraint and states that the total number of nodes visited by the source cannot exceed n (n depends on the fuel capacity of the source). Constraint (14) defines the multistatic detection criteria. Constraint (15) is a technical constraint that defines the relationship between decision variables z and x together with Constraint (16). Constraint (17) is a technical constraint that ensures that z equals 1 for the starting node $g^{ss}$. Constraint (18) is a subtour elimination constraint.

## 5. ACO Implementation

Although the mathematical model yields an optimal solution for the problem, the calculation time extends in large-scale grids. Despite this challenge, a viable solution can be attained by using a metaheuristic approach.

ACO is a metaheuristic approach for combinatorial optimization problems which has been structured by [29,30]. ACO simulates the pheromone-laying behavior of ants, where artificial ants traverse a problem space to find optimal solutions by depositing and following pheromone trails. Over the years, researchers have developed various heuristic strategies to enhance ACO’s performance in diverse problem domains, such as routing, scheduling, and resource allocation.

The pheromone mechanism is an artificial value in order to simulate ants’ pheromones, where a trail is constructed for the other ants in the nest to reach a food source. In ACO, a pheromone value is assigned to each feasible solution element. More formally, the pheromone value $\tau_{ij}$ is assigned to each feasible solution element $c_{ij}$, which comprises the appointment $D_{i}=v_{i}^{j}$, where $D_{i}$ is a finite set of decision variables and $v_{i}^{j}$ is the assigned value of the decision variable.

During the execution of ACO, a simulated virtual ant creates a solution by exploring an entirely connected graph $G_{c}\left(V,E\right)$, where V is a set of vertices and E is a set of edges. The aforementioned graph may be fetched from the elements of a solution set C by the vertices or edges depiction. Ants traverse between vertices through edges and gradually construct a fractional solution. Furthermore, ants leave a specific amount of pheromone Δτ while traversing on the solution components, either vertices or edges, depending on the implemented solution technique. This amount is based on the level of solution created. The following ants utilize this information as a reference to be directed to the prospective part of the search area.

The ACO heuristic implementation is designed to compute heuristic values for each hexagonal node in the grid, guiding the movement of ants in the ACO algorithm. The heuristic function integrates the covering criteria of each node. This heuristic strategy creates a guidance system where ants are primarily driven toward the target but are also influenced by the nodes with a higher coverage ratio.

Our heuristic implementation initializes the transition values for each hexagonal node within the ACO framework. Transition values represent the probability or desirability for an ant to move from one hexagonal node to another during the search process. This implementation establishes the initial transition value for a given hexagonal node based on its spatial relationship with both other hexagons and receiver nodes. For each hexagonal node in the grid, the distance to both the target node and the nearest receiver node is calculated. This condition reflects a proximity-based weighting mechanism and is based on multistatic detection criteria. If the product of the two distances is less than or equal to the square of the ρ value, the transition value is incremented. This strategy ensures that hexagonal nodes that have higher coverage values receive higher initial transition values, reinforcing their attractiveness as viable paths for the ants. The combination of proximity-based weighting and threshold-based selection allows the ACO algorithm to balance exploration and exploitation in the hexagonal grid environment.

The implementation dynamically updates the transition values of hexagonal cells based on the ant’s performance and trail. This mechanism forms the core of the pheromone update process in the ACO framework, allowing the algorithm to reinforce successful paths and adapt the search strategy over successive iterations. This step models the natural decay of pheromones over time, preventing premature convergence and ensuring that the search process remains dynamic. A higher evaporation rate accelerates the decay, encouraging exploration, while a lower rate promotes the retention of learned paths.

The implementation computes the transition probabilities that guide the movement of ants between hexagonal nodes during the search process. This method is a key component of the ACO framework, where the transition probability is influenced by both pheromone levels and heuristic information. The calculated probabilities determine the likelihood of selecting a particular neighboring hexagon as the next step in the path. The process begins by identifying the neighboring hexagonal nodes of the ant’s current position. For each valid neighboring hexagonal node that has not been visited by the ant, the method computes a transition probability based on the pheromone level on the hexagonal node and the heuristic value. If all computed probabilities are zero (e.g., when the pheromone levels are very low or the heuristic values are unfavorable), the implementation assigns equal transition probabilities to all valid neighboring nodes. This ensures that the ant is not trapped and retains the ability to explore other paths.

The ACO heuristic implementation evaluates the coverage achieved by an ant at each step during its search process. Coverage in the context of hexagonal nodes refers to the number of hexagons influenced or visited by the ant based on its position and proximity to receivers. The goal of this method is to measure how much of the grid is being explored by the ant and how well the ant’s path interacts with key points of interest (i.e., receivers). This evaluation helps to refine the ant’s search behavior by reinforcing paths that maximize coverage and exploring areas that contribute to the overall solution quality. The product of distances approach creates a Cassini oval coverage zone around each hexagon, influenced by both the proximity to the ant and the nearest receiver. The parameter ρ acts as a control for adjusting the effective radius of influence, where a larger value of ρ increases the potential coverage zone.

The pseudo-code of our ACO heuristic implementation is presented in Algorithm 1.
**Algorithm 1** ACO Implementation1:Initialize hexCenterCoordList, receiverCenterCoordList, startCenterCoord, endCenterCord, maxIterations, evaporationRate, destinationWeight, receiverDistanceWeight, randomFactor2:Compute grid dimensions based on hexagon side length and area size.3:Initialize pheromone values and store them in hexTransitionValues.4:Create ant population of size numberOfAnts5:**for each** hexagon in hexCenterCoordList **do**6:    Compute the coverage status of each hexagon.7:    Store these values in hexHeuristicValues.8:**end for**9:**for each** hexagon in hexCenterCoordList **do**10:    Identify neighbor hexagons.11:    **for each** neighbor hexagon **do**12:        Calculate the distance product between each receiver and neighbor hexagon.13:        **if** this product is within the square of the range of the day value **then**14:            Increase the transition value.15:**            break**16:**        end if**17:**    end for**18:**end for**19:**for each** attempt **do**20:    Reset ant positions and trails.21:    Initialize pheromone levels.22:    **while** the iteration count is less than maxIterations **and** no ant has reached the endCenterCoord **do**23:        **for each** ant **do**24:            Identify neighbor hexagons.25:            Remove visited hexagons from the candidate list.26:            **if** random probability is less than randomFactor **then**27:                Pick a random neighbor.28:**            else**29:                Calculate the transition probability for each valid neighbor30:                Select the next move based on cumulative probability distribution.31:**            end if**32:**        end for**33:        **for each** hexagon in hexCenterCoordList **do**34:            Evaporate the current pheromone level based on evaporationRate.35:**        end for**36:        **for each** ant reaching endCenterCoord within maxIterations **do**37:            Increase pheromone values on their trails proportional to the contribution.38:
        **end for**
39:
    **end while**
40:    **if** no ant reaches endCenterCoord within maxIterations **then**41:        Select the ant with the highest contribution as the fallback solution.42:
    **end if**
43:    Display the best path visualization on the hexagonal grid.44:**end for**

## 6. Numerical Results

### 6.1. Experiment Settings

We now assess the performance of the MILP model and ACO implementation in terms of solution quality and computational durations in terms of CPU time. To achieve this, we generate four different problem sizes in two dimensions. Next, we generate 10 random replications for each problem size with each fixed parameter combination. In total, 90 × 4 = 360 replications are solved with the proposed MILP model. The average of each 10 runs is noted as the final result. Each of the problem instances is solved with an optimality gap setting of 0% with a CPU time limit of one hour. In this configuration, the run will conclude either when a feasible solution is shown to fall within the designated gap of the optimal objective function value or when the CPU time resource limit is exceeded.

To facilitate the analysis of the MILP model, we utilize a method, which is implemented in MATLAB R2023b 64-bit (win64), to randomly generate problem instances for each dimensional option. Receiver locations are generated according to a uniform distribution randomly across all problem sizes. For each problem instance, we create a hexagonal-grid-based area incorporating receiver locations, as well as start and end nodes, along with a coverage matrix for all locations on the hexagonal grid by using this method. Distances between nodes are calculated as Euclidean distance. The implementation of the MILP model is carried out in GAMS Release 24.0.1 and is solved with CPLEX 12.5.0.0 for x86_64/MS Windows and Gurobi Library Version 5.0.2.

The same problem instance replications with the same parameter settings are used in ACO implementation, which is implemented with the Java programming language. We run the code 10 times for each replication and take the average of the solutions as the final result for that group.

All computations are carried out on a system equipped with an 11th Gen Intel(R) Core(TM) i7-11800H @ 2.30 GHz CPU and 24 GB of RAM.

We present the variables for all the problem sizes used in our numerical experiments in Table 1. The table includes two parts from top to bottom. The first part of the table includes the size-related parameters, particularly the area length, node length, calculated circumradius, and node numbers. The largest size S4 has 304 nodes, while the smallest size S1 has 42 nodes. The second part of the table shows the values of fixed parameters for the problem instances to create different combinations of the problem sizes.

The optimality gap values in both the mathematical model and the ACO solution are calculated according to the following formulas:(19)$$optcr=optca/\left(10^{-9}+max\left(abs\left(EstObjVal_{MATMOD}\right),\;abs\left(ResObjVal_{MATMOD\;or\;ACO}\right)\right)\right)$$
where optca represents the absolute optimality criterion of the mathematical model, optcr represents the relative optimality criterion of the mathematical model and ACO implementation, $EstObjVal_{MATMOD}$ denotes the estimated objective value of the mathematical model, and $ResObjVal_{MATMOD\;or\;ACO}$ represents the resulting objective value found by the mathematical model and ACO implementation. The optcr value is used as the optimality gap result for both the mathematical model and ACO implementation runs.

### 6.2. Results and Discussion

In this section, we present the results of our model and ACO solution experiments. We show the results of the proposed model and solution with respect to solution quality and computation time. In Table 2, we report the performance of the model and the solution with respect to different sizes and parameter configurations, averaged over 10 replications for each problem size. Values from columns 4 and 5 represent the number of problem instances that returned an integer solution within the predefined optimality gap setting of 0% and their average CPU time. Values from columns 6 and 7 represent the number of problem instances that returned a feasible integer solution within the pre-specified time limit of 1 h and the average optimality gap to optimality. Columns 8 and 9 include the average computational time in seconds and the average optimality gaps for the problem instances of the ACO solution.

The results reveal that for the optimality gap setting 0%, CPLEX can solve approximately 50.83% of the instances (183 out of 360) to optimality with an average CPU time of 104.792 s and can find a feasible integer solution for 49.17% of the instances (177 out of 360) with an average of 20.4% optimality gap. The maximum optimality gap is observed with S4 problems with four receivers and a ρ value of 3. Overall, the findings demonstrate that our formulation based on the hexagonal decomposition approach provides an effective framework for small to moderately sized instances, delivering optimal or near-optimal solutions with reasonable computational effort. However, as the problem size increases, as expected we observe increased solution times and higher relative optimality gaps.

The key observation is the behavior of the relative difference between the exact solver’s and ACO’s optimality gaps. For small-size problem instances of S1 and S2 types, ACO outperformed the model in one case and found solutions with less than 7% difference from the model’s solutions. For the S3 type of problem, ACO found the optimal solutions where the model also reached the optimum and outperformed or showed less than 1% difference from the model’s solutions. For the largest problem instances of the S4 type, ACO outperformed the solver in all cases. This trend suggests that as problem complexity increases, ACO does not deteriorate compared to the model’s performance but leads to better solutions.

Another key advantage of the ACO heuristic is its efficiency in computational time. The maximum average CPU time for the heuristic implementation is 1543.278 s for S4 type problem instances with three receivers and rho value of 6. In this case, the mathematical model finds a feasible integer solution within 1 h with an average optimality gap of 40.93%. But our ACO-based heuristic approach reaches a feasible integer solution with an optimality gap of 34.50% within 1543.278 s.

## 7. Conclusions

MSNs have been extensively utilized in underwater surveillance operations. Although such operations are conducted widely, the theoretical background for their practical utilization is an active research field. Since multistatic operations with mobile sonar are complex because of their unconventional coverage zones, the task of efficient route planning is vital.

The goal of this work is to provide a theoretical approach to implementing an efficient solution for providing optimal routes for a mobile source in an MSN. Although there exist several studies related to area coverage maximization or cost reduction for stationary bistatic or multistatic sonar systems in the literature, to the best of our knowledge, this is the first study to propose a solution for generating an optimal route for a mobile source in an MSN system.

Our MILP formulation employs a cookie-cutter (deterministic) sensing model and utilizes a hexagonal-grid-based map to maximize the number of hexagonal nodes covered in this map. Although it demonstrates good performance in small-sized problem instances, the solution times increase significantly in large instances. The binary nature of the cookie-cutter model, while simplifying the formulation, limits realism by ignoring uncertainties such as detection probability, noise, and false alarms. As a future study, we will consider more sophisticated, probabilistic sensing models to account for this uncertainty.

The comparison between the commercial solver and ACO heuristic reveals certain strengths and weaknesses. While the solver ensures optimality for certain problem instances, its computational burden limits its applicability to larger problem instances. ACO, though suboptimal, offers significant computational advantages, making it a suitable approach for large-scale problem-solving. Notably, the decreasing relative difference in optimality gaps as problem size increases suggests that ACO becomes a more viable heuristic in complex scenarios.

While fixed receiver positions and uniform environmental conditions simplify modeling in MSNs, they introduce limitations that may impact real-world applicability. Accounting for the changes in receiver locations and environmental variability through adaptive methodologies can enhance the robustness of both mathematical models and heuristic solutions. Additionally, the current framework assumes static point coverage during planning, which simplifies the problem but does not fully capture the challenges of real-world underwater surveillance scenarios. Target mobility could significantly impact optimal paths and detection opportunities, requiring real-time route adjustments and predictive models to maintain coverage effectiveness. Future work will incorporate dynamic target behavior to better align the system with operational realities.

Another future study could consider evaluating the model’s integration with practical autonomous platforms to test operational feasibility in order to satisfy modern underwater surveillance operations. Although comparisons with alternative coverage optimization methods are beyond the current scope of this study, they can be addressed in subsequent studies. We believe that the hexagonal grid structure is particularly well suited for area coverage scenarios, as it reduces coverage gaps, provides uniform connectivity, and ensures better area representation compared to traditional square grids. Our ACO implementation’s performance can be further improved through hybrid optimization techniques or adaptive parameter tuning strategies. Although our approach was inspired by underwater surveillance operations and submerged sensing systems, it can be extended for air and land surveillance frameworks.

This research is confined to the domain of mathematical modeling and optimization for sensor network navigation in underwater environments. The study’s core objective is to enhance the efficiency of sonar sensor usage in complex spatial domains, which is of interest in academic fields such as operations research, control systems, and marine robotics. Although the techniques may be adaptable to military scenarios, we emphasize that no classified systems, technologies, or operational doctrines are used or revealed. The results are generic and methodological in nature, and all experiments are performed in a simulated environment using abstract sensor models that reflect general principles of sonar coverage rather than real-world classified parameters. The data and models used do not include any restricted military data or simulation of specific operational scenarios. We acknowledge the potential dual-use nature of this work and confirm that all necessary precautions have been taken to prevent misuse. As part of our ethical responsibility, we adhere strictly to national and international guidelines concerning dual-use research. We advocate for transparency, regulatory compliance, and responsible application to ensure the research serves peaceful and socially beneficial purposes.

## Figures and Tables

**Figure 1 sensors-25-04139-f001:**
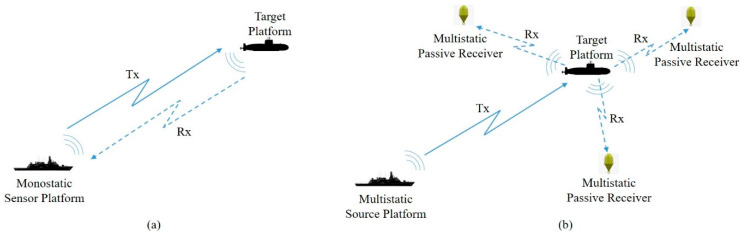
(**a**) A monostatic sonar system where receiving and transmitting units are integrated together; (**b**) a multistatic sonar system where transmitting and three receiving units are positioned separately.

**Figure 2 sensors-25-04139-f002:**
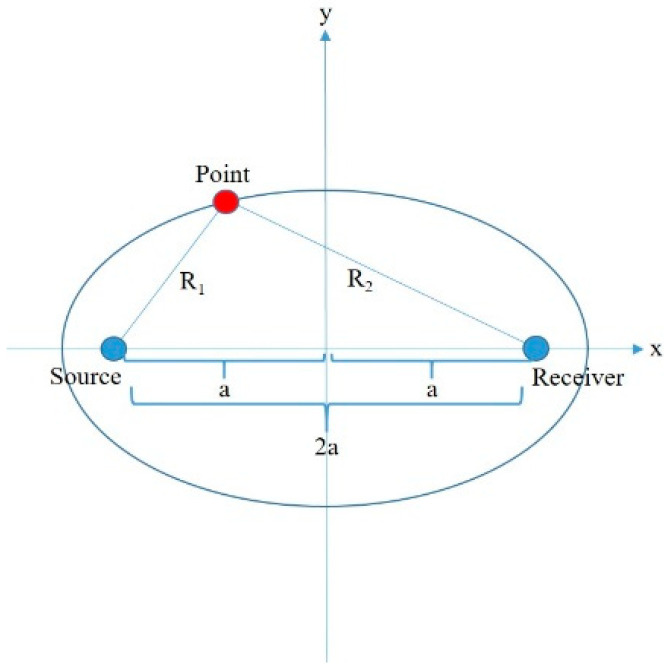
Bistatic triangle.

**Figure 3 sensors-25-04139-f003:**
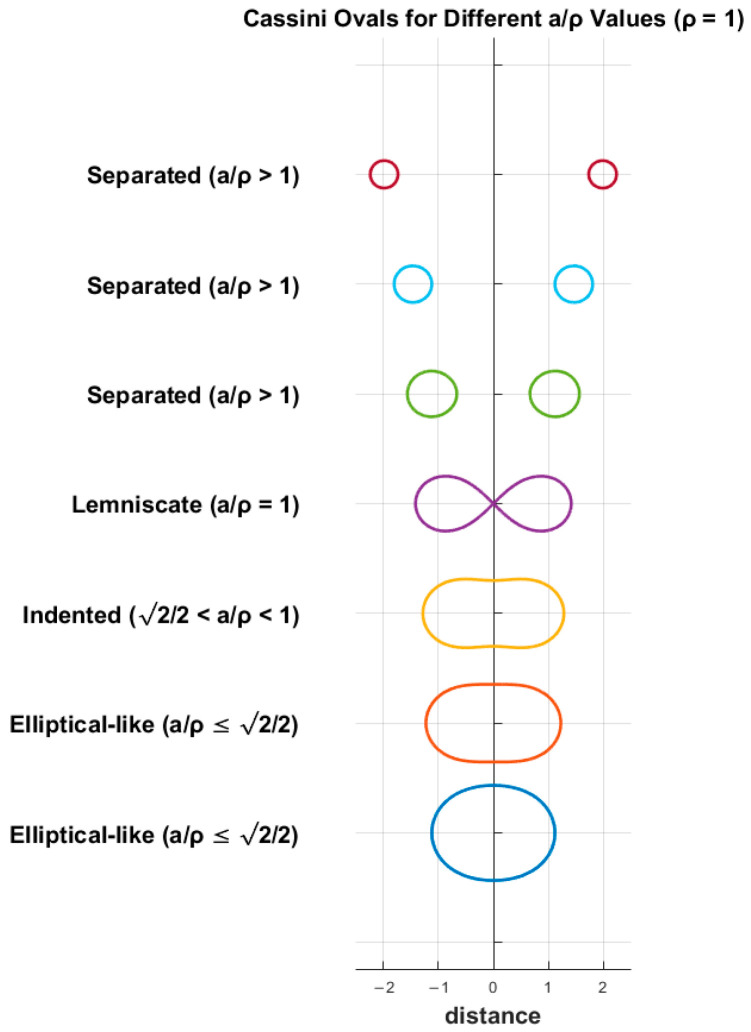
Cassini ovals for ρ=1 and various separation distances.

**Figure 4 sensors-25-04139-f004:**
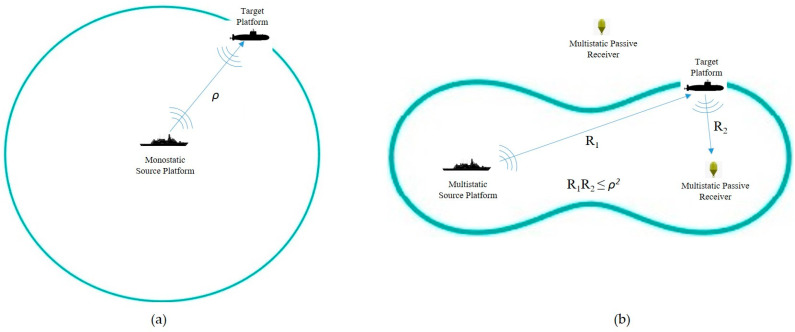
(**a**) A monostatic sonar system with an enclosed loop sensing range of radius ρ; (**b**) a multistatic sonar formation with a singular transmitting unit and receiving unit.

**Table 1 sensors-25-04139-t001:** Problem sizes.

		Problem Size			
		S1	S2	S3	S4
Size-Related Parameters	Area Length	10 × 10	15 × 15	20 × 20	25 × 25
	Hex Length	1	1	1	1
	Hex Circumradius	0.866	0.866	0.866	0.866
	Number of Nodes	42	105	188	304
Fixed Parameters	ρ Value	1, 1.5, 2	1, 2, 3	1, 2, 3	1, 2, 3
	# of Receivers	2, 4, 6	2, 4, 6	2, 4, 6	2, 4, 6
	Max. # of Nodes Visited	10	20	25	30

**Table 2 sensors-25-04139-t002:** Performance of the proposed mathematical model and ACO solution.

		Mathematical Model					ACO Heuristic	
Problem Size	ρ Value	# of Receivers	# of Optimum Solutions	Average CPU Time (s)	Number of Feasible Integer Solutions	Average Optimality Gap	Average CPU Time (s)	Average Optimality Gap
S1	1	2	10	0.096	0	0%	20.1793	0%
		4	10	0.147	0	0%	30.0559	0%
		6	10	0.083	0	0%	37.2324	0%
	1.5	2	10	0.634	0	0%	19.6918	7.06%
		4	10	1.511	0	0%	29.548	1.53%
		6	10	1.23	0	0%	75.3542	4.02%
	2	2	10	1.118	0	0%	22.123	1.73%
		4	10	5.915	0	0%	35.512	1.02%
		6	10	12.985	0	0%	50.4965	1.49%
S2	1	2	10	9.384	0	0%	18.9668	0%
		4	10	8.168	0	0%	32.2091	0%
		6	10	10.672	0	0%	45.4981	0%
	2	2	5	41.3	5	7.56%	24.4027	6.37%
		4	0	-	10	6.65%	52.9074	8.6%
		6	0	-	10	14.79%	73.0621	17.55%
	3	2	0	-	10	17.43%	36.537	17.71%
		4	0	-	10	16.46%	64.901	18.55%
		6	0	-	10	16.58%	127.0262	18.59%
S3	1	2	10	113.697	0	0%	87.4682	0%
		4	10	223.601	0	0%	158.8933	0%
		6	10	117.434	0	0%	206.4308	0%
	2	2	1	126.2	9	7.74%	107.512	8.02%
		4	0	-	10	19.83%	241.084	19.05%
		6	0	-	10	21.88%	324.8215	21.46%
	3	2	0	-	10	21.66%	140.9292	21.49%
		4	0	-	10	26.88%	351.984	27.38%
		6	0	-	10	29.64%	552.1855	30.19%
S4	1	2	9	566.148	1	3.13%	261.4247	0%
		4	10	431.653	0	0%	491.945	0%
		6	9	423.869	1	2.78%	769.7606	0%
	2	2	0	-	10	14.71%	287.1949	12.96%
		4	0	-	10	27.93%	636.4105	22.54%
		6	0	-	10	32.97%	1053.662	27.54%
	3	2	0	-	10	33.27%	406.5589	26.45%
		4	0	-	10	45.17%	1030.250	37.51%
		6	0	-	10	40.93%	1543.278	34.50%

## Data Availability

Data set (except the source code) available on request from the authors.

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
