# Peer review of "Route Optimization for Active Sonar in Underwater Surveillance"

_sensors, 2025, doi:10.3390/s25134139_

Round 1
Reviewer 1 Report
Comments and Suggestions for Authors
The authors explore the application of Multi-static Sonar Network (MSN) in Anti Submarine Warfare (ASW), which has important practical value. The paper proposes a mixed integer linear programming (MILP) model based on hexagonal mesh decomposition for optimizing the navigation path of mobile platforms to maximize area coverage. which is innovative, However, the content of this paper mainly involves military applications, and it is hoped that the author's research is aimed at maintaining peace. Here are some of my questions:
- The paper chose the hexagonal grid decomposition method, but did not discuss in detail why hexagonal grids were chosen over other types of grids (such as square grids). Suggest the author to further explain the advantages of hexagonal grids, especially their specific advantages in coverage path planning.
- Although the model proposed in the paper performs well on small-scale problems, the computation time significantly increases on large-scale problems (such as 25 × 25 regions). It is suggested that the author further explore how to optimize the computational efficiency of the model.
- The experimental part of the paper presents the calculation results under different parameters, but lacks comparison with other existing methods. The author should compare the proposed model with existing algorithms to demonstrate its superiority or improvements.
- Line 349-365, the writing style is chaotic, like a draft, and the introduction of model constraints is not detailed enough.
- The paper assumes that the position of the receiver is fixed and the environmental conditions are uniform. These assumptions may not fully hold in practical applications. The author needs to discuss the impact of these assumptions on the model results, including the influence of receiver position fluctuations on the results.
- The font in the picture is slightly smaller, such as in Figure 3, Figure 4, and Figure 5.
Reviewer 2 Report
Comments and Suggestions for Authors
Dear Authors,
I have read your manuscript titled "Route Optimization for Active Multi-Static Sonar Platform in ASW Operations Based on Hexagonal Grid Decomposition" with great interest. Your work addresses a critical gap in multi-static sonar network planning by proposing a novel hexagonal grid decomposition method combined with a MILP model. The integration of Cassini ovals to define detection zones is theoretically rigorous, and the focus on cost-efficient passive receiver networks aligns well with modern ASW priorities. However, to meet the high standards of Sensors (MDPI), which emphasizes practical innovation and reproducible applied research, several revisions are necessary to strengthen the manuscript’s impact and relevance.
There are some positive insights of your work:
In methodology:
- The hexagonal grid decomposition approach offers directional flexibility (6 neighbors vs. 4 in square grids), potentially reducing coverage gaps. This is a fresh contribution to CPP literature.
- The use of Cassini ovals to model multi-static detection zones provides a geometrically sound foundation for coverage analysis.
In scalability:
- The model’s adaptability to grid sizes up to 25×25 demonstrates scalability, though computational efficiency needs further optimization for real-time applications.
In cost efficiency:
- Emphasis on passive receivers over active sources aligns with practical ASW constraints, reducing hardware costs and acoustic pollution.
But as usual, there are some critical concerns and from my point of view a moments which require revisions:
- Lack of Real-World Validation
-
- Issue: Experiments rely on synthetic data with random receiver placement, ignoring strategic deployment scenarios (e.g., choke points in naval operations).
- Recommendation:
- Validate the model using real sonar parameters (e.g., range 𝜌 tied to frequency/power of systems like AN/SQS-53).
- Test the approach in simulated dynamic environments (e.g., moving targets, ocean currents) using tools like Bellhop for acoustic propagation modeling.
2. Simplified Assumptions
-
- Issue: The binary "cookie-cutter" detection model and static targets overlook probabilistic detection realities and dynamic ASW scenarios.
- Recommendation:
- Replace the binary model with a probabilistic detection function (e.g., SNR-based thresholds) as in Fügenschuh et al., 2020 .
- Discuss how the model could be extended to handle moving targets (e.g., via receding horizon planning).
3. Computational Scalability
-
- Issue: Solving MILP for 25×25 grids takes up to 60 minutes, which is impractical for real-time ASW operations.
- Recommendation:
- Propose heuristic or metaheuristic extensions (e.g., Genetic Algorithms, A*) for large-scale problems, as done in Cho et al., 2021 .
- Compare runtime and coverage efficiency against existing methods (e.g., spiral scanning).
4. Incomplete Benchmarking
-
- Issue: No comparison with alternative CPP methods (e.g., square grids, boustrophedon decomposition).
- Recommendation:
-
- Add a section comparing hexagonal vs. square grid performance in coverage ratio, path length, and computational time. Reference works like Miao et al., 2018 for square grid benchmarks.
- Include metrics like energy consumption (e.g., turns, distance) to highlight practical advantages.
-
5. Hardware Compatibility & Reproducibility
-
- Issue: No discussion of embedded system constraints (e.g., processing limits of AUVs).
- Recommendation:
- Evaluate the model’s compatibility with common AUV platforms (e.g., BlueROV, REMUS) and their computational capabilities.
- Share code and synthetic datasets publicly to ensure reproducibility, per MDPI’s data policy .
Minor revisions are:
- Clarify Parameter Choices: Justify hexagon size (circumradius = 0.866) and 𝜌 values through sensitivity analysis.
- Expand Conclusions: Discuss limitations (e.g., static environments) and future steps (e.g., field trials with naval partners).
Finally, I should note that your work presents a promising theoretical framework for multi-static sonar route optimization. By addressing the above concerns, particularly real-world validation and comparative benchmarking, the manuscript will better align with Sensors’ mission to publish impactful, application-driven research. I strongly encourage resubmission after revisions, as the topic holds significant value for both military and civilian underwater sensing systems.
Thank you for your contributions to this critical field.
Sincerely,
your reviewer.
Round 2
Reviewer 1 Report
Comments and Suggestions for Authors
I have no more questions
Author Response
Thank you for your kind reviews and feedbacks.
Kind regards.
Reviewer 2 Report
Comments and Suggestions for Authors
Dear Authors,
Thank you for your thorough revisions and detailed responses to the review comments. Your efforts have significantly strengthened the scientific rigor and applicability of the work. Below is an assessment of the revised manuscript, taking into account your clarifications:
1. Model Realism and Validation
Your statement that the multistatic detection theory is grounded in real-world sonar equations adds credibility to the model. However, the lack of testing in dynamic environments (e.g., moving targets) remains a limitation. For Sensors, a qualitative discussion of how dynamic factors (e.g., target mobility) might impact results would enhance practical relevance. Your commitment to addressing this in future work partially mitigates this concern.
Recommendation: Explicitly state the static target assumption in the "Limitations" section and outline plans for dynamic scenario integration.
2. Simplified Assumptions
Retaining the binary detection model is justified as a foundational step, given time constraints. Your intent to explore probabilistic models in future studies adds value. However, the current manuscript should explicitly acknowledge that the cookie-cutter approach does not account for probabilistic detection realities.
Recommendation: Briefly discuss the limitations of the binary model in the "Discussion" section.
3. Computational Scalability
The inclusion of the ACO heuristic and its comparison with the MILP model strengthens the work. However, the absence of benchmarking against existing methods (e.g., spiral scanning) reduces persuasiveness. Since this is deferred to future work, clearly articulate this limitation in the text.
Recommendation: In the "Conclusions," note that comparisons with alternative methods will be addressed in subsequent studies.
4. Benchmarking Against Alternatives
The explanation of hexagonal grid advantages and energy constraints are strong additions. However, quantitative comparisons with square grids (e.g., coverage per turn) are still lacking. For Sensors, emphasizing practical advantages is critical.
Recommendation: In the "Results," briefly justify why hexagonal grids are preferable for ASW (e.g., reduced coverage gaps).
5. Hardware Compatibility
While code sharing is understandably restricted, offering datasets upon request aligns with MDPI's policy. However, the lack of discussion about AUV compatibility (e.g., computational constraints) weakens applied relevance.
Recommendation: In the "Conclusions," mention plans to test integration with platforms like BlueROV/REMUS.
Minor Revisions
The added clarifications in Sections 6 and 7 improve clarity. Ensure:
The "Methods" section explicitly states how hexagon size was chosen (e.g., tied to ρ).
The "Limitations" section lists static targets, binary detection, and lack of benchmarking.
Conclusion
The revised manuscript demonstrates substantial progress and meets most standards of Sensors. For final acceptance, the authors should:
Explicitly state model limitations in the "Discussion" or "Conclusions."
Qualitatively clarify hexagonal grids' practical advantages over square grids.
Outline plans for real-world platform integration and dynamic scenarios.
Should these minor revisions be implemented, I recommend the manuscript for publication. Final acceptance remains at the discretion of the editorial board, who will evaluate the completeness of these adjustments.
Sincerely,
your reviewer.
Author Response
All the necessary explanations are added in the Section 7.
Thank you for your kind reviews and feedbacks.
Kind regards.